# ORTHOGONAL SELF-ATTENTION

## ABSTRACT

Softmax Self-Attention (SSA) is a key component of Transformer architectures. However, when utilised within skipless architectures, which aim to improve representation learning, recent work has highlighted the inherent instability of SSA due to inducing rank collapse and poorly-conditioned Jacobians. In this work, we design a novel attention mechanism: Orthogonal Self-Attention (OSA), which aims to bypass these issues with SSA, in order to allow for (non-causal) Transformers without skip connections and normalisation layers to be more easily trained. In particular, OSA parametrises the attention matrix to be orthogonal via mapping a skew-symmetric matrix, formed from query-key values, through the matrix exponential. We show that this can be practically implemented, by exploiting the low-rank structure of our query-key values, resulting in the computational complexity and memory cost of OSA scaling linearly with sequence length. Furthermore, we derive an initialisation scheme for which we prove ensures that the Jacobian of OSA is well-conditioned.

## 1 INTRODUCTION

Skip connections (He et al., 2016) have become an ubiquitous feature of neural network architectures from facilitating the stable training of deep models. However, despite their success, recent works (Veit et al., 2016; Gromov et al., 2024; Zhang et al., 2024) have raised the concern that the benefits of skip connections, namely ease of training, may be obscuring deeper issues, in terms of representation learning, that skip connections induce. The main point behind these criticisms is that skip connections appear to bias models away from properly utilising the full depth of their architectures. For instance, Ji et al. (2025a) argues that since skip connections continually reintroduce earlier features into deeper layers, they disrupt the learning of hierarchical and progressively more abstract representations, fundamentally harming representation learning.

Motivated by this line of reasoning, we explore designing Transformers that are able to be trained stably without skip connections. Prior works (He et al., 2023; Ji et al., 2025a) have tackled this through modifications to Softmax Self-Attention (SSA) (Vaswani et al., 2017) and weight initialisations to improve signal propagation and the conditioning of the Jacobian matrix. However, these works restrict themselves to standard Softmax-based Transformers which appear to be inherently unstable without skip connections (Dong et al., 2021; Ji et al., 2025b) due to SSA.

Therefore, due to the fundamental issues with SSA, in this work, we propose *Orthogonal Self-Attention (OSA)* which attempts to circumvent the training instability of skipless SSA by designing the attention matrix to be orthogonal. This is motivated by the rank collapse phenomenon associated with SSA (Dong et al., 2021; Noci et al., 2022) where token representations quickly converge to a rank-1 matrix with depth. In contrast, by enforcing the attention matrix to be orthogonal in OSA, we preserve the rank of representation, which should mitigate against rank collapse in skipless architectures.

In terms of how we implement OSA, we parametrise the attention matrix via the matrix exponential, mapping skew-symmetric matrices, computed from query-key values, to the manifold of (special) orthogonal matrices. To make the use of the matrix exponential tractable in practice, we present a scheme for efficiently computing the matrix exponential through exploiting our low-rank design of the skew-symmetric matrices. This allows OSA to scale linearly, in terms of computational complexity and memory cost, with sequence length, in contrast to the quadratic scaling that SSA

requires. We note that this does restricts the applicability of OSA to non-causal decoder-based Transformers such as ViTs (Dosovitskiy, 2020) and DiTs (Peebles & Xie, 2023).

Finally, we analyse how OSA impacts the conditioning of the network Jacobian. To make our analysis tractable, we use the same assumptions as employed in Ji et al. (2025b;a), which reduces the analysis of the conditioning of the network Jacobian to the conditioning of the individual attention sub-blocks. Motived by these assumptions, we derive an initialisation scheme for OSA for which we prove ensures that the Jacobian of OSA (with respect to the input) is well-conditioned. We provide an empirical validation of our initialisation scheme and OSA by replacing the SSA layers of a ViT with OSA and benchmarking this novel model on MNIST (LeCun et al., 2002) for classification.

## 2 ORTHOGONAL SELF-ATTENTION

In this work, we consider non-causal decoder-based Transformers (e.g. ViTs and DiTs) where we will replace the use of SSA with Orthogonal Self-Attention (OSA) and remove skip connections and the use of normalisation layers.

Let $\mathbf{X}_0 \in \mathbb{R}^{N \times d}$ be the initial token representations computed from the input to the architecture, where $N$ denotes the number of tokens and $d$ denotes the representation dimension. We define an *OSA-Transformer* by the following recursion:

$$\hat{\mathbf{X}}_l = \text{M-OSA}(\mathbf{X}_{l-1}) \tag{1}$$

$$\mathbf{X}_l = \text{MLP}(\hat{\mathbf{X}}_l) \tag{2}$$

where $\mathbf{X}_l$ denotes the token representations after the $l$-th transformer block and MLP denotes some MLP applied token-wise. Furthermore, we define M-OSA as the multihead version of OSA defined as

$$\text{M-OSA}(\mathbf{X}_{l-1}) = \sum_{i=1}^{h} \text{OSA}_{l,i}(\mathbf{X}_{l-1}) \tag{3}$$

$$= \sum_{i=1}^{h} \mathbf{A}_i(\mathbf{X}_{l-1})\mathbf{X}_{l-1}\mathbf{W}_{l,i}^V\mathbf{W}_{l,i}^O \tag{4}$$

where $h$ is the number of heads, $\text{OSA}_{l,i}$ denotes OSA for a single head, $\mathbf{A}_i \in \text{SO}(N) \subset \mathbb{R}^{N \times N}$ is an attention matrix, $\mathbf{W}_{l,i}^V \in \mathbb{R}^{d \times d_v}$ and $\mathbf{W}_{i,l}^O \in \mathbb{R}^{d_v \times d}$ are the value and output weights respectively (where $d_v = \frac{d}{h}$), and $i$ indexes the heads. For simplicity, we will consider $\text{OSA}_{l,i}$ for a single layer and head, and we will drop the indices $l$ and $i$ for the remainder of the paper.

Finally, we define OSA for some input $\mathbf{X} \in \mathbb{R}^{N \times d}$ as

$$\text{OSA}(\mathbf{X}) = \mathbf{A}(\mathbf{X})\mathbf{X}\mathbf{W}^V\mathbf{W}^O, \text{ where } \mathbf{A}(\mathbf{X}) = \exp(\mathbf{S}) \text{ and } \mathbf{S} = \frac{\alpha}{\sqrt{d_v}}\left(\mathbf{Q}\mathbf{K}^\top - \mathbf{K}\mathbf{Q}^\top\right), \tag{5}$$

where $\exp$ denotes the matrix exponential (Hall, 2013), $\alpha \in \mathbb{R}$ is some scalar learnable parameter, and $\mathbf{Q} = \mathbf{X}\mathbf{W}^Q, \mathbf{K} = \mathbf{X}\mathbf{W}^K \in \mathbb{R}^{N \times d_v}$ are the query, key matrices with the respective weights $\mathbf{W}^Q, \mathbf{W}^K \in \mathbb{R}^{d \times d_v}$.

We note that $\mathbf{S}$ is defined to be skew-symmetric which ensures that $\mathbf{A}(\mathbf{X})$ is a (special) orthogonal matrix, we use the scaling $\frac{1}{\sqrt{d_v}}$ for normalising the dot-product of vectors in $\mathbb{R}^{d_v}$, and we include $\alpha$ to aid with our initialisation scheme we define later on. Furthermore, it is also easy to see that OSA is permutation equivariant with respect to the token positions.

### 2.1 IMPLEMENTATION DETAILS

We note that a naive implementation of OSA has computational complexity of $O(N^3)$ due to the matrix exponential. However, Theorem 2.1 shows that we can greatly reduce this through exploiting the low-rank structure of $\mathbf{S}$, as we usually have that $d_v$ is small compared to $N$ and $\mathbf{S}$ has $r \leq 2d_v$ where $r = \text{rank } \mathbf{S}$. For the proof, see Appendix F.1.

**Theorem 2.1.** *Let* $\mathbf{B}(\mathbf{X}) \in \mathbb{R}^{N \times r}$ *be a matrix where the columns provide an orthonormal basis for the subspace* $U$ *spanned by the columns of* $\mathbf{Q}, \mathbf{K}$. *Then we have:*

$$\exp(\mathbf{S}(\mathbf{X})) = \mathbf{I}_N + \mathbf{B}(\mathbf{X})\left(\exp[\mathbf{S}](\mathbf{X}) - \mathbf{I}_r\right)\mathbf{B}(\mathbf{X})^\top, \tag{6}$$

*where*

$$[\mathbf{S}](\mathbf{X}) = \mathbf{B}(\mathbf{X})^\top \mathbf{S}(\mathbf{X})\mathbf{B}(\mathbf{X}) \in \mathbb{R}^{r \times r}. \tag{7}$$

This reduces the problem of computing $\exp(\mathbf{S})$ to computing $\exp[\mathbf{S}]$ which has a computational complexity of $O(r^3) \ll O(N^3)$.

## 2.2 Kernel Analysis

The following theorem shows that OSA does not suffer from the rank collapse issue associated with SSA. For the proof, see Appendix F.2

**Theorem 2.2.** *Consider a skipless OSA-only Transformer (i.e. without MLP blocks) with* $h = 1$ *at initialisation where we initialise* $\mathbf{W}_l^V \mathbf{W}_l^O \in \mathbb{R}^{d \times d}$ *to be orthogonal. Let* $\mathbf{X}_l$ *be the output after the* $l$-*th layer, then the layer-wise kernel matrix* $\Sigma_l = \mathbf{X}_l \mathbf{X}_l^\top$ *has the form:*

$$\Sigma_l = \mathbf{A}\Sigma_0 \mathbf{A}^\top, \tag{8}$$

*where* $\mathbf{A} \in \mathrm{SO}(N)$ *is some orthogonal matrix. Therefore, the rank and eigenvalues of* $\Sigma_0$ *are preserved.*

## 2.3 Basis Computation

In order to be able to apply Theorem 2.1, we need to be able to construct $\mathbf{B}(\mathbf{X})$. The standard approach to achieve this is to apply a reduced QR decomposition to the matrix $\mathbf{M} = [\mathbf{Q}, \mathbf{K}] \in \mathbb{R}^{N \times 2d_v}$ where we usually have $2d_v < N$. However, this can suffer from exploding gradients when $\mathbf{M}$ is close to being rank deficient, and induces a bias on the ordering of column due to the fact that QR always fixes the first column of $\mathbf{M}$ (up to some scaling) when constructing $\mathbf{B}(\mathbf{X})$ (Roberts & Roberts, 2020).

Alternatively, we can compute $\mathbf{B}(\mathbf{X})$ via the Newton-Schulz iterates (Higham, 2008):

$$\mathbf{B}(\mathbf{X}) = \mathbf{M}_K, \text{ where } \mathbf{M}_{k+1} = \frac{1}{2}\mathbf{M}_k(3\mathbf{I}_{2d_v} - \mathbf{M}_k^\top \mathbf{M}_k) \text{ and } \mathbf{M}_0 = \mathbf{M}, \tag{9}$$

where $K$ is some fixed choice of iterations. When $\mathbf{M}$ is full rank and the singular values $\{\sigma_i(\mathbf{M})\}_i$ of $\mathbf{M}$ satisfy $\sigma_i(\mathbf{M}) < \sqrt{3}$ for all $i$, $\mathbf{M}_K$ converges towards the matrix $\mathbf{U}$ of orthonormal basis elements given by the Polar decomposition of $\mathbf{M}$. Due to the requirement on singular values and the inequality $\sigma_{\max}(\mathbf{M}) \leq \|\mathbf{M}\|_F$, where $\sigma_{\max}$ denotes the maximum singular value and $\|\cdot\|_F$ denotes the Frobenius norm, we follow standard practice by pre-normalising the matrix $\mathbf{M}$ so that $\mathbf{M}_0 = \frac{\mathbf{M}}{\|\mathbf{M}\|_F + \epsilon}$ where $\epsilon > 0$ is for numerical stability. Further, we note that there exists further efficiency gains that could be explored, from recent work such as (Amsel et al., 2025), as well as noticing that we can unroll the repeated application of Newton-Schulz iterates to express $\mathbf{M}_K$ in terms of $\mathbf{M}h(\mathbf{M}^\top \mathbf{M})$ where $h$ is some polynomial and $\mathbf{M}^\top \mathbf{M} \in \mathbb{R}^{2d_v \times 2d_v}$ in order to reduce the computational complexity required. We also provide an erorr

We note that this approach has the nice properties that due to the process only requiring matrix multiplications, it is stable when $\mathbf{M}$ is close to being rank deficient and tends to be more efficient than QR from more effective hardware utilization. Moreover, $\mathbf{U}$ is not biased towards any feature dimensions (we can view $\mathbf{U}$ as the closest orthonormal matrix under the Frobenius norm to $\mathbf{M}$).

## 2.4 Complexity and Memory Analysis

For simplicity, we assume that we have $h = 1$ and $r = 2d$. It is fairly easy to show that by exploiting the low-rank structure of $\mathbf{S}$, the computational complexity of OSA is $O(Nd^2 + d^3)$ and the memory cost is $O(Nd + d^2)$ which scales linearly with $N$. In contrast, the computational complexity of SSA is $O(N^2 d)$ and the memory cost is $O(N^2)$. We provide a more detailed breakdown of our derivation in Appendix B.

## 2.5 Initialisation for OSA

Due to space constraints, we refer the reader to Appendix C for the initialisation scheme we use for OSA and the Jacobian analysis.

## 3 Experiments

In this section, we provide an initial validation of OSA. We consider a standard ViT architecture for classification from Dosovitskiy (2020) on MNIST (LeCun et al., 2002). We then design an OSA-Transformer by taking this ViT architecture and replacing SSA with OSA, as well as removing skip connections and layer norm (LN) (Ba et al., 2016) from the rest of the model and keeping everything else the same; we also apply the initialisation scheme from Section C.

In Figure 1, we provide a comparison of train and test loss curves for our OSA-Transformer and ViT baseline. We note that for all models, we use AdamW (Loshchilov & Hutter, 2017) and the same training hyper-parameters (such as learning rate, weight decay etc.). For the ViT baseline, we ablate removing skip connections and removing both skip connections and layer norm, and for our OSA-Transformer, we ablate the use of QR and Newton-Schultz in OSA. For further details and results, see Appendix E.

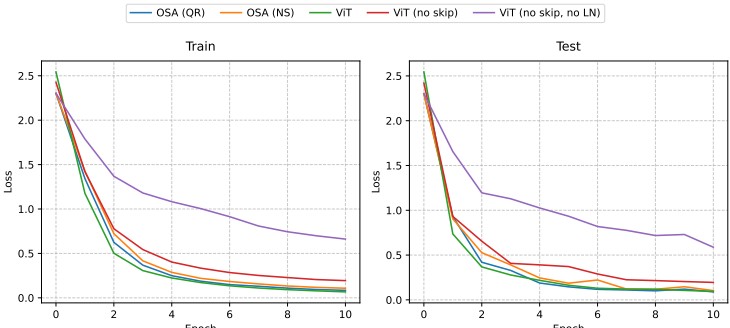

Figure 1: Train and test loss curves for OSA-Transformer and ViT models trained on MNIST classification.

For the ViT models, we see that removing skip connections and layer norm results in worse training speed and generalisation, with the removal of layer norm especially degrading performance. As for the OSA-Transformer models, we interestingly see that *despite* removing skip connections and layer norm, our model is able to match the generalisation performance of the ViT baseline and approaches its training speed, outperforming the ViT without skip connections[1]. Additionally, we see that QR appears to slightly outperform Newton-Schultz, and we note that we found QR to be numerically stable during training.

While these results are not extensive, they help to suggest that the design and proper initialisation of OSA can result in Transformer-based architectures that are able to be trained efficiently, even without the use of skip connections and normalisation layers.

## 4 Conclusion

In this work, we have introduced Orthogonal Self-Attention (OSA), which aims to circumvent the poor performance of SSA-based Transformers when skip connections and normalisation layers are removed, by parametrising the attention matrix to be orthogonal via the matrix exponential function. In the future, we will look to expand on the empirical validation of OSA, as well as investigating the potential benefits that OSA might provide, in terms of representation learning, from allowing models to be trained without skip connections or normalisation layers.

---

[1]We expect for more complex dataset that the performance difference between ViT and ViT (no skip) will be larger (He et al., 2023; Ji et al., 2025a).

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

## A    EXTRA RESULTS ON BASIS COMPUTATION

One thing to be careful of when using Newton-Schultz is that the matrix $\mathbf{B}$ is not guaranteed to provide an orthonormal basis when the algorithm has not converged, thus the right-hand side of Equation 6 may not be exactly orthogonal. In Theorem A.1 we provide a bound for how this "orthogonality error" depends on how orthonormal $\mathbf{B}$ is—i.e. the convergence of Newton-Schultz. For the proof, see Appendix F.3.

**Theorem A.1.** *Let* $\mathbf{Y} = \mathbf{I}_N + \mathbf{B}(\mathbf{X})\left(\exp[\mathbf{S}](\mathbf{X}) - \mathbf{I}_r\right)\mathbf{B}(\mathbf{X})^\top$ *and let* $\{\sigma_i(\mathbf{B})\}_i$ *denote the singular values of* $\mathbf{B}$. *We have the following bound:*

$$\|\mathbf{Y}^\top\mathbf{Y} - \mathbf{I}_N\|_2 \leq \left(e^{\|\mathbf{S}\|_2} - 1\right)^2 \max_i |\sigma_i(\mathbf{B})^2(\sigma_i(\mathbf{B})^2 - 1)| \tag{10}$$

$$\leq \frac{1}{4}\left(e^{\|\mathbf{S}\|_2} - 1\right)^2, \tag{11}$$

*where* $\|\cdot\|_2$ *denotes the spectral norm.*

Interestingly, the above bound shows that the orthogonality error is robust to very small singular values of $\mathbf{B}$ and that larger rotations incur a higher error if $\mathbf{B}$ has not converged well (of course this effect is offset if $\mathbf{B}$ has converged which suggests an interesting trade-off).

## B    EXTRA DETAILS ON COMPLEXITY AND MEMORY ANALYSIS

In Table 1, we provide a breakdown of the computational complexity and memory cost of different components within OSA.

Table 1: Computational and memory complexity of OSA components.

| Computation | Complexity | Memory |
|---|---|---|
| **Reduced QR on** $\mathbf{M}$ | $O(Nd^2)$ | $O(Nd)$ |
| **Newton-Schultz on** $\mathbf{M}$ | $O(Nd^2)$ | $O(Nd)$ |
| **Computing** $[\mathbf{S}]$ | $O(Nd^2 + d^3)$ | $O(Nd)$ |
| **Computing** $\exp[\mathbf{S}]$ | $O(d^3)$ | $O(d^2)$ |
| **Computing Eq (6) and applying** $\mathbf{X}\mathbf{W}^V\mathbf{W}^O$ | $O(Nd^2 + d^3)$ | $O(Nd)$ |

## C    EXTRA DETAILS ON INITIALISATION FOR OSA

We analyse the Jacobian of OSA to propose an initialisation scheme that ensures that the Jacobian is well-conditioned at initialisation. For simplicity, we only consider a single head throughout this section. Following the justification of Ji et al. (2025b;a)[2], it is reasonable to assume that this will improve the trainability and performance of an OSA-Transformer. In Theorem C.1, we provide the form of the Jacobian. For the proof, see Appendix F.4.

**Theorem C.1.** *The Jacobian* $\mathbf{J} \in \mathbb{R}^{Nd \times Nd}$ *of* OSA *with respect to the input* $\mathbf{X}$ *is given by*

$$\mathbf{J}(\mathbf{X}) = \frac{\partial \operatorname{vec}\operatorname{OSA}(\mathbf{X})}{\partial \operatorname{vec}\mathbf{X}} \tag{12}$$

$$= \underbrace{(\mathbf{X}\mathbf{W}^V\mathbf{W}^O \otimes \mathbf{I}_N)^\top \frac{\partial \operatorname{vec}\mathbf{A}(\mathbf{X})}{\partial \operatorname{vec}\mathbf{X}}}_{\mathbf{J}_1} + \underbrace{(\mathbf{W}^V\mathbf{W}^O)^\top \otimes \mathbf{A}(\mathbf{X})}_{\mathbf{J}_2}, \tag{13}$$

*where* vec *denotes the operator that converts a matrix to its (column-dominant) vectorised form and* $\otimes$ *denotes the Kronecker product.*

---

[2]i.e. the conditioning of the full network Jacobian is bounded by the worse-conditioned sub-block and that the attention sub-blocks are much worse conditioned than the MLP sub-blocks.

## C.1 Value-Output Initialisation

We see that a natural desideratum to ensure $\mathbf{J}$ is well-conditioned is to enforce $\kappa(\mathbf{W}^V \mathbf{W}^O) = 1$ where $\kappa(\mathbf{D}) = \frac{\sigma_{\max}(\mathbf{D})}{\sigma_{\min>0}(\mathbf{D})}$ denotes the effective condition number of the matrix $\mathbf{D}$ and $\sigma_{\min>0}$ denotes the minimum non-zero singular value[3]. This can be achieved by sampling $\mathbf{U}, \mathbf{V} \overset{\text{i.i.d.}}{\sim} \mathcal{U}_{d \times d_v}$ and setting $\mathbf{W}^V = \mathbf{U}, \mathbf{W}^O = \mathbf{V}^\top$, where $\mathcal{U}_{n \times m}$ denotes the uniform distribution over the Stiefel manifold $V_m(\mathbb{R}^n) = \{\mathbf{W} \in \mathbb{R}^{n \times m} : \mathbf{W}^\top \mathbf{W} = \mathbf{I}_m\}$. For details on sampling from $\mathcal{U}_{n \times m}$, see Appendix D.

## C.2 Query-Key Initialisation

From Equation 13, we note that the query, key weights only affect the Jacobian via the term: $\tilde{\mathbf{W}} = \mathbf{W}^Q (\mathbf{W}^K)^\top - \mathbf{W}^K (\mathbf{W}^Q)^\top$. To see this in more detail, we refer to the proof of Theorem C.3 which shows that the Jacobian of $\mathbf{A}(\mathbf{X})$ is a function of $\tilde{\mathbf{W}}$. Therefore, to help ensure all terms in the Jacobian are well-conditioned, we aim to enforce $\kappa(\tilde{\mathbf{W}}) = 1$. We can achieve this by sampling $\mathbf{U} \sim \mathcal{U}_{d \times 2d_v}$ and setting $\mathbf{W}^Q = \mathbf{U}_1, \mathbf{W}^K = \mathbf{U}_2$ where $\mathbf{U} = [\mathbf{U}_1, \mathbf{U}_2]$. It is easy to see that this scheme requires $2d_v \le d$ which is satisfied for $h > 1$. We prove in Theorem C.2 that this satisfies our requirement. For the proof, see Appendix F.5.

**Theorem C.2.** *We assume $2d_v \le d$. If we initialise $\mathbf{W}^Q, \mathbf{W}^K$ such that $[\mathbf{W}^Q, \mathbf{W}^K] \in \mathbb{R}^{d \times 2d_v}$ forms an orthonormal matrix, we have*

$$\kappa(\tilde{\mathbf{W}}) = 1. \tag{14}$$

*Specifically, we have that the non-zero singular values of $\tilde{\mathbf{W}}$ are 1.*

## C.3 $\alpha$ Initialisation and Jacobian Analysis

Our analysis of the conditioning of the Jacobian $\mathbf{J}$ is complicated by the fact that the term $\mathbf{J}_1$ is hard to reason about. On the other hand, our initialisation scheme has been designed to ensure the condition number of $\mathbf{J}_2$ is very close to (or exactly) 1. Therefore, one strategy is to show that we can control the spectral norm of $\mathbf{J}_1$, so we can model this term as a small perturbation which leaves the singular values of $\mathbf{J}_2$ mostly intact. In Theorem C.3, we show that we have linear control over $\|\mathbf{J}_1\|_2$ in terms of $\alpha$. For the proof, see Appendix F.6.

**Theorem C.3.** *Let $\mathbf{J}_1(\mathbf{X}) = (\mathbf{X} \mathbf{W}^V \mathbf{W}^O \otimes \mathbf{I}_N)^\top \frac{\partial \operatorname{vec} \mathbf{A}(\mathbf{X})}{\partial \operatorname{vec} \mathbf{X}}$. We assume $\mathbf{W}^Q, \mathbf{W}^K, \mathbf{W}^V, \mathbf{W}^O$ follow our above initialisation, the spectral norm of $\mathbf{X}$ is bounded and we use Newton-Schultz to compute $\mathbf{B}$. We then have the following bound:*

$$\|\mathbf{J}_1(\mathbf{X})\|_2 \le C\alpha, \tag{15}$$

*where $C > 0$ is some constant.*

This provides the motivation to complete our initialisation scheme where we initialise $\alpha$ to be some small number. Indeed this can viewed as setting $\mathbf{A}(\mathbf{X})$ to be close to the identity matrix at the start of training.

As a consequence of the above bound, we have Theorem C.4 which shows under our initialisation (assuming Newton-Schultz converges well enough), we can set the condition number of the Jacobian to be arbitrary close to 1. For the proof, see Appendix F.7.

**Theorem C.4.** *Under the same assumptions as Theorem C.3, we further assume that $|\sigma_i(\mathbf{A}) - 1| \le \delta$[4] for some $\delta \in [0, 1)$. We have the following bound:*

$$1 \le \kappa(\mathbf{J}(\mathbf{X})) \le \frac{1 + \delta + C\alpha}{1 - \delta - C\alpha}, \tag{16}$$

*for any $\alpha > 0$ such that $1 - \delta - C_\alpha > 0$.*

---

[3]We consider this version of the condition number as $\mathbf{W}^V \mathbf{W}^O$ and other matrices we consider are constrained to be low rank since we consider the case where $h > 1$.

[4]If we use QR for OSA, we have $\sigma_i(\mathbf{A}) = 1$ for all $i$. We use this assumption for the case where we use Newton-Schultz with finite $K$ so that $\mathbf{A}$ is not exactly an orthogonal matrix. We see from Theorem A.1 that this orthogonality error is fairly robust to Newton-Schultz not fully converging.

**Remark C.5.** We note that a similar result should be available for the case when $\mathbf{B}$ is computed using the QR decomposition but would require more assumptions on the properties of $\mathbf{M}$.

This bound suggests that our initialisation scheme, provided we set $\alpha$ to be small enough and $K$ large enough, ensures that the Jacobian of OSA is well-conditioned at initialisation.

### C.4 MLP INITIALISATION

The focus of this paper has been on the design of OSA and its initialisation as there already exists extensive work on the design and initialisation of skipless MLPs. In particular, we use the SUO initialisation scheme from Martens et al. (2021) for the MLP components of an OSA-Transformer.

## D SAMPLING

We provide pseudocode for sampling from $\mathcal{U}_{n \times m}$ in Algorithm 1. For further details on this algorithm, we refer to Section 4 and 5 in Mezzadri (2006)[5].

---

**Algorithm 1** Sampling from $\mathcal{U}_{n \times m}$

---

**Require:** Dimensions $n, m$ where $n \geq m$
1: Sample random matrix $\mathbf{Z} \in \mathbb{R}^{n \times m}$ with entries $\mathbf{Z}_{ij} \overset{\text{i.i.d.}}{\sim} \mathcal{N}(0, 1)$
2: Compute reduced QR decomposition: $\mathbf{Z} = \mathbf{QR}$ where $\mathbf{Q} \in \mathbb{R}^{n \times m}, \mathbf{R} \in \mathbb{R}^{m \times m}$
3: **for** $i \leftarrow 1$ to $m$ **do**
4: $\quad s_i \leftarrow \text{sign}(\mathbf{R}_{ii})$
5: **end for**
6: $\mathbf{Q} \leftarrow \mathbf{Q} \cdot \text{diag}(s_1, \ldots, s_m)$           ▷ Multiplies $i$-th column of $\mathbf{Q}$ by $s_i$
7: **return** $\mathbf{Q}$

---

## E EXPERIMENTAL DETAILS

For the ViT architecture, we take the standard design from Dosovitskiy (2020). For the embedding layer, we use a patch size of 4, we add the `[cls]` token for classification and we have learnable 1D position embeddings. This results in $n = 50$ as images from MNIST are $28 \times 28$. We use 6 Transformer blocks and we take $d = 64$. For SSA and OSA, we take $h = 4$ and for the MLPs, we have a width ratio of 4 and we use GELU activations (Hendrycks, 2016); OSA also initialises $\alpha$ to be 0.1 and when using Newton-Schultz, we use $K = 6$. For the output layer, we take the representation of the `[cls]` token and project it to logits using a linear layer. We do not use dropout in any models.

For training, we train for 10 epochs with a batch size of 128, learning rate of 3e-4, weight decay of 0.05, gradient norm clipping of 1.0. We also use the standard cross-entropy loss and AdamW for training.

For the initialisation of weights in the ViT models, we use Xavier uniform initialisation for the weights of SSA and MLP and we initialise bias terms to zero. For the learnable embeddings and `[cls]` representation, we use the truncated Gaussian initialisation given by $\mathcal{N}(0, \sigma^2)\mathbf{1}_{[-2\sigma, 2\sigma]}$ ($\sigma = 0.02$).

For OSA-Transformer, we use the initialisation scheme from Section C. We note that we also initialise the bias terms of the MLP layers to zero and we do not use bias terms for any of the weights in the OSA layers. For the embedding layer and output layer, we use the same initialisation as the ViT models.

For further experimental results, in Table 2, we provide the final test accuracy of models after training.

---

[5]This reference focuses on the case where $n = m$, however, to sample from $\mathcal{U}_{n \times m}$, we can equivalently sample from $\mathcal{U}_{n \times n}$ first and then truncate the matrix.

Table 2: Test accuracy for OSA-Transformer and ViT models trained on MNIST classification after 10 epochs.

| Model | OSA (QR) | OSA (NS) | ViT | ViT (no skip) | ViT (no skip, no LN) |
|---|---|---|---|---|---|
| **Test Accuracy** | 97.2% | 96.9% | 97.2% | 94.6% | 79.6% |

# F PROOFS

## F.1 PROOF OF THEOREM 2.1

*Proof.* Firstly, let $U$ be the subspace given by the image of $\mathbf{S}$. This is spanned by the span of the column space of $\mathbf{Q}$ and $\mathbf{K}$. Moreover, by the design of $\mathbf{S}$, we can view $\mathbf{S}$ as a linear operator defined on the subspace $U$ ($\dim U = r$) instead of the space $\mathbb{R}^N$. To show this, take $x \in U^\perp$, then $\mathbf{S}x = \frac{\alpha}{\sqrt{d_v}}\left(\mathbf{Q}\mathbf{K}^\top - \mathbf{K}\mathbf{Q}^\top\right)x$. By definition, $x$ is orthogonal to the columns of both $\mathbf{Q}, \mathbf{K}$, hence $\mathbf{S}x = 0$. Therefore, $\mathbf{S}$ can be considered as a linear mapping $U \to U$.

Secondly, to represent $\mathbf{S}$ in terms of a $\mathbb{R}^{r \times r}$ matrix, we take an orthonormal basis $u_1, \ldots, u_r$ of $U$ and define $\mathbf{B} \in \mathbb{R}^{N \times r}$ by $\mathbf{B} = [u_1, \ldots, u_r]$. We note that $\mathbf{B}^\top : \mathbb{R}^N \to \mathbb{R}^r$ is a change-of-basis map sending a vector in $U$ to its coordinate representation given by $u_1, \ldots, u_r$ and that $\mathbf{B} : \mathbb{R}^r \to \mathbb{R}^N$ is the inverse mapping and $\mathbf{B}^\top \mathbf{B} = \mathbf{I}_r$. From this, we can conclude that there exists a matrix $[\mathbf{S}] = \mathbf{B}^\top \mathbf{S} \mathbf{B} \in \mathbb{R}^{r \times r}$ that represents $\mathbf{S}$ in the coordinates given by $u_1, \ldots, u_r$, hence, we have

$$\mathbf{S} = \mathbf{B}[\mathbf{S}]\mathbf{B}^\top \implies \mathbf{S}^n = \mathbf{B}[\mathbf{S}]^n\mathbf{B}^\top. \tag{17}$$

Therefore, we have

$$\exp(\mathbf{S}) = \sum_{n=0}^\infty \frac{\mathbf{S}^n}{n!} = \mathbf{I}_N + \sum_{n=1}^\infty \frac{\mathbf{S}^n}{n!} = \mathbf{I}_N + \sum_{n=1}^\infty \frac{\mathbf{B}[\mathbf{S}]^n\mathbf{B}^\top}{n!} = \mathbf{I}_N + \mathbf{B}(\exp[\mathbf{S}] - \mathbf{I}_r)\mathbf{B}^\top. \tag{18}$$

$\square$

## F.2 PROOF OF THEOREM 2.2

*Proof.* It is trivial to show that:

$$\mathbf{X}_l = \mathbf{A}\mathbf{X}_0\mathbf{W}, \tag{19}$$

where $\mathbf{A} = \mathbf{A}_l \ldots \mathbf{A}_1$ and $\mathbf{A}_l$ is the attention matrix computed at the $l$-th layer, and $\mathbf{W} = \mathbf{W}_1^V\mathbf{W}_1^O \ldots \mathbf{W}_l^V\mathbf{W}_l^O$. As we enforce that each attention matrix is orthogonal, we have that $\mathbf{A} \in \mathrm{SO}(N)$, and from our initialisation, we have that $\mathbf{W}\mathbf{W}^\top = \mathbf{I}_d$. This allows us to conclude the result. $\square$

## F.3 PROOF OF THEOREM A.1

*Proof.* Let $\mathbf{E} = \mathbf{B}^\top\mathbf{B} - \mathbf{I}_{2d_v}$ and $\Delta = \exp[\mathbf{S}] - \mathbf{I}_{2d_v}$, then $\mathbf{Y} = \mathbf{I}_N + \mathbf{B}\Delta\mathbf{B}^\top$. We first note the identity:

$$\Delta^\top\Delta = (\exp[\mathbf{S}] - \mathbf{I}_{2d_v})^\top(\exp[\mathbf{S}] - \mathbf{I}_{2d_v}) \tag{20}$$
$$= 2\mathbf{I}_{2d_v} - \exp[\mathbf{S}] - \exp[\mathbf{S}]^\top \tag{21}$$
$$= -\Delta - \Delta^\top. \tag{22}$$

We now expand $\mathbf{Y}^\top\mathbf{Y} - \mathbf{I}_N$:

$$\mathbf{Y}^\top\mathbf{Y} - \mathbf{I}_N = (\mathbf{I}_N + \mathbf{B}\Delta\mathbf{B}^\top)^\top(\mathbf{I}_N + \mathbf{B}\Delta\mathbf{B}^\top) - \mathbf{I}_N \tag{23}$$
$$= \mathbf{B}\Delta\mathbf{B}^\top + \mathbf{B}\Delta^\top\mathbf{B}^\top + \mathbf{B}\Delta^\top\mathbf{B}^\top\mathbf{B}\Delta\mathbf{B}^\top \tag{24}$$
$$= \mathbf{B}\Delta\mathbf{B}^\top + \mathbf{B}\Delta^\top\mathbf{B}^\top + \mathbf{B}\Delta^\top(\mathbf{E} + \mathbf{I}_{2d_v})\Delta\mathbf{B}^\top \tag{25}$$
$$= \mathbf{B}\Delta\mathbf{B}^\top + \mathbf{B}\Delta^\top\mathbf{B} + \mathbf{B}\Delta^\top\mathbf{E}\Delta\mathbf{B}^\top + \mathbf{B}\Delta^\top\Delta\mathbf{B}^\top \tag{26}$$
$$= \mathbf{B}\left(\Delta + \Delta^\top + \Delta^\top\mathbf{E}\Delta + \Delta^\top\Delta\right)\mathbf{B}^\top \tag{27}$$
$$= \mathbf{B}\Delta^\top\mathbf{E}\Delta\mathbf{B}^\top, \tag{28}$$

where we apply our identity. Next, let $\mathbf{B} = \mathbf{U}\Sigma\mathbf{V}^\top$ be the SVD decomposition of $\mathbf{B}$ where $\mathbf{U} \in \mathbb{R}^{N \times N}, \Sigma \in \mathbb{R}^{N \times 2d_v}, \mathbf{V} \in \mathbb{R}^{2d_v \times 2d_v}$. Using this decomposition, we have

$$\Delta = \exp(\mathbf{B}^\top \mathbf{S} \mathbf{B}) - \mathbf{I}_{2d_v} \tag{29}$$

$$= \exp(\mathbf{V}\Sigma^\top \mathbf{U}^\top \mathbf{S} \mathbf{U}\Sigma\mathbf{V}^\top) - \mathbf{I}_{2d_v} \tag{30}$$

$$= \mathbf{V}\hat{\Delta}\mathbf{V}^\top, \tag{31}$$

where $\hat{\Delta} = \exp([\hat{\mathbf{S}}]) - \mathbf{I}_{2d_v}$ and $[\hat{\mathbf{S}}] = \Sigma^\top \mathbf{U}^\top \mathbf{S} \mathbf{U}\Sigma$, and

$$\mathbf{E} = \mathbf{B}^\top \mathbf{B} - \mathbf{I}_{2d_v} \tag{32}$$

$$= \mathbf{V}\Sigma^\top \mathbf{U}^\top \mathbf{U}\Sigma\mathbf{V}^\top - \mathbf{I}_{2d_v} \tag{33}$$

$$= \mathbf{V}\hat{\mathbf{E}}\mathbf{V}^\top, \tag{34}$$

where $\hat{\mathbf{E}} = \Sigma^\top \Sigma - \mathbf{I}_{2d_v}$. Further, applying this to Equation 28, we have

$$\mathbf{Y}^\top \mathbf{Y} - \mathbf{I}_N = \mathbf{U}\Sigma\hat{\Delta}^\top \hat{\mathbf{E}}\hat{\Delta}\Sigma^\top \mathbf{U}^\top. \tag{35}$$

Next, we note we have the following decomposition of $\hat{\Delta}$:

$$\hat{\Delta} = \exp(\Sigma^\top \mathbf{U}^\top \mathbf{S} \mathbf{U}\Sigma) - \mathbf{I}_{2dv} \tag{36}$$

$$= \sum_{n=1}^{\infty} \frac{1}{n!}(\Sigma^\top \mathbf{U}^\top \mathbf{S} \mathbf{U}\Sigma)^n \tag{37}$$

$$= \Sigma^\top \left( \underbrace{\sum_{n=1}^{\infty} \frac{1}{n!}(\mathbf{U}^\top \mathbf{S} \mathbf{U}\Sigma\Sigma^\top)^{n-1}\mathbf{U}^\top \mathbf{S} \mathbf{U}}_{\Psi} \right) \Sigma, \tag{38}$$

which allows us to rewrite Equation 35 as

$$\mathbf{Y}^\top \mathbf{Y} - \mathbf{I}_N = \mathbf{U}\Sigma\Sigma^\top \Psi^\top \Sigma\hat{\mathbf{E}}\Sigma^\top \Psi\Sigma\Sigma^\top \mathbf{U}^\top. \tag{39}$$

This implies the following bound:

$$\|\mathbf{Y}^\top \mathbf{Y} - \mathbf{I}_N\|_2 \leq \|\mathbf{U}\|_2^2 \|\Sigma\|_2^4 \|\Psi\|_2^2 \|\Sigma(\Sigma^\top \Sigma - \mathbf{I}_{2d_v})\Sigma^\top\|_2 \tag{40}$$

$$\leq \|\Psi\|_2^2 \|\Sigma(\Sigma^\top \Sigma - \mathbf{I}_{2dv})\Sigma^\top\|_2, \tag{41}$$

where we note that $\|\Sigma\|_2 \leq 1$ as we normalise the singular values of $\mathbf{M}_0$ to be less than 1 and the Newton-Schultz iterations cannot increase initial singular values in the range $(0, 1]$ to be more than 1.

To analyse $\|\Psi\|_2$, we have the following bound:

$$\|\Psi\|_2 \leq \sum_{n=1}^{\infty} \frac{1}{n!} \|\mathbf{U}^\top \mathbf{S} \mathbf{U}\Sigma\Sigma^\top\|_2^{n-1} \|\mathbf{U}^\top \mathbf{S} \mathbf{U}\|_2 \tag{42}$$

$$\leq \sum_{n=1}^{\infty} \frac{\|\mathbf{S}\|_2^n}{n!} \tag{43}$$

$$= e^{\|\mathbf{S}\|_2} - 1. \tag{44}$$

To analyse $\|\Sigma(\Sigma^\top \Sigma - \mathbf{I}_{2dv})\Sigma^\top\|_2$, we note that $\Sigma$ has a diagonal form with non-zero values given by $\sigma_i(\mathbf{B})$ which are the singular values of $\mathbf{B}$. Therefore, the singular values of the matrix $\Sigma(\Sigma^\top \Sigma - \mathbf{I}_{2dv})\Sigma^\top$ have the form: $|\sigma_i(\mathbf{B})^2(\sigma_i(\mathbf{B})^2 - 1)|$, hence

$$\|\Sigma(\Sigma^\top \Sigma - \mathbf{I}_{2dv})\Sigma^\top\|_2 = \max_i |\sigma_i(\mathbf{B})^2(\sigma_i(\mathbf{B})^2 - 1)|. \tag{45}$$

We note that by construction $\sigma_i(\mathbf{B}) \in [0, 1]$ therefore the maximum possible value of the above expression is $\frac{1}{4}$ (from inspecting the graph of $|x^2(x^2 - 1)|$).

Putting all of this together, we get the final result:

$$\|\mathbf{Y}^\top \mathbf{Y} - \mathbf{I}_N\|_2 \leq \left( e^{\|\mathbf{S}\|_2} - 1 \right)^2 \max_i |\sigma_i(\mathbf{B})^2(\sigma_i(\mathbf{B})^2 - 1)|. \tag{46}$$

$\square$

### F.4    PROOF OF THEOREM C.1

*Proof.* This is a standard application of matrix differentials (Magnus & Neudecker, 2019). By the product rule of matrix differentials, we have

$$\partial\,\mathrm{OSA}(\mathbf{X}) = \partial\mathbf{A}(\mathbf{X})(\mathbf{X}\mathbf{W}^V\mathbf{W}^O) + \mathbf{A}(\mathbf{X})\partial\mathbf{X}(\mathbf{W}^V\mathbf{W}^O) + \mathbf{A}(\mathbf{X})\mathbf{X}\partial(\mathbf{W}^V\mathbf{W}^O). \tag{47}$$

By the identity, $(A \otimes B)\,\mathrm{vec}\,C = \mathrm{vec}(BCA^\top)$, we have

$$\partial\,\mathrm{vec}\,\mathrm{OSA}(\mathbf{X}) = ((\mathbf{X}\mathbf{W}^V\mathbf{W}^O)^\top \otimes \mathbf{I}_N)\partial\,\mathrm{vec}\,\mathbf{A}(\mathbf{X}) + ((\mathbf{W}^V\mathbf{W}^O)^\top \otimes \mathbf{A}(\mathbf{X}))\partial\,\mathrm{vec}\,\mathbf{X} \tag{48}$$

$$+ (\mathbf{I}_{d_v} \otimes \mathbf{A}(\mathbf{X}))\partial\,\mathrm{vec}(\mathbf{W}^V\mathbf{W}^O). \tag{49}$$

We note that $\partial\,\mathrm{vec}\,\mathbf{X}/\partial\,\mathrm{vec}\,\mathbf{X} = \mathbf{I}$ and $\partial\,\mathrm{vec}(\mathbf{W}^V\mathbf{W}^O)/\partial\,\mathrm{vec}\,\mathbf{X} = 0$. Therefore, we can conclude the following form:

$$\frac{\partial\,\mathrm{vec}\,\mathrm{OSA}(\mathbf{X})}{\partial\,\mathrm{vec}\,\mathbf{X}} = (\mathbf{X}\mathbf{W}^V\mathbf{W}^O \otimes \mathbf{I}_N)^\top \frac{\partial\,\mathrm{vec}\,\mathbf{A}(\mathbf{X})}{\partial\,\mathrm{vec}\,\mathbf{X}} + (\mathbf{W}^V\mathbf{W}^O)^\top \otimes \mathbf{A}(\mathbf{X}). \tag{50}$$

□

### F.5    PROOF OF THEOREM C.2

*Proof.* We have $\tilde{\mathbf{W}} = \mathbf{W}^Q(\mathbf{W}^K)^\top - \mathbf{W}^K(\mathbf{W}^Q)^\top$. We consider $\tilde{\mathbf{W}}\tilde{\mathbf{W}}^\top$:

$$\tilde{\mathbf{W}}\tilde{\mathbf{W}}^\top = -\tilde{\mathbf{W}}^2 \tag{51}$$

$$= -\left(\mathbf{W}^Q(\mathbf{W}^K)^\top - \mathbf{W}^K(\mathbf{W}^Q)^\top\right)\left(\mathbf{W}^Q(\mathbf{W}^K)^\top - \mathbf{W}^K(\mathbf{W}^Q)^\top\right) \tag{52}$$

$$= \mathbf{W}^Q(\mathbf{W}^Q)^\top + \mathbf{W}^K(\mathbf{W}^K)^\top, \tag{53}$$

where we use the fact that $\tilde{\mathbf{W}}$ is skew-symmetric and $(\mathbf{W}^Q)^\top\mathbf{W}^Q = (\mathbf{W}^K)^\top\mathbf{W}^K = \mathbf{I}$ and $(\mathbf{W}^Q)^\top\mathbf{W}^K = 0$. We note that the right-hand side is a projection matrix onto the subspace spanned by $\mathbf{W}^Q, \mathbf{W}^K$ therefore the non-zero eigenvalues of $\tilde{\mathbf{W}}\tilde{\mathbf{W}}^\top$ are all 1. This implies the result as the singular values of $\tilde{\mathbf{W}}$ are the square-root of the absolute value of the eigenvalues of $\tilde{\mathbf{W}}\tilde{\mathbf{W}}^\top$. □

### F.6    PROOF OF THEOREM C.3

To organise our proof, we present the following lemmas. We leave the original assumptions in the statement of the theorem implicit.

**Lemma F.1.** *There exists some constant $C > 0$ such that*

$$\|\mathbf{B}\|_2 \leq C. \tag{54}$$

*Proof.* When using Newton-Schultz, we pre-normalise our matrix $\mathbf{M}$ so that the singular values are less than 1 and we note that Newton-Schultz cannot increase the singular values greater than $\sqrt{3}$ for all choices of $K$[6]. □

**Lemma F.2.** *There exists some constant $C > 0$ such that*

$$\|\mathbf{S}\|_2 \leq C\alpha. \tag{55}$$

*Proof.* To analyse $\mathbf{S}$, we let $\tilde{\mathbf{W}} = \mathbf{W}^Q(\mathbf{W}^K)^\top - \mathbf{W}^K(\mathbf{W}^Q)^\top$. We can write $\mathbf{S} = \frac{\alpha}{\sqrt{d_v}}\mathbf{X}\tilde{\mathbf{W}}\mathbf{X}^\top$ which provides the bound

$$\|\mathbf{S}\|_2 \leq \frac{\alpha}{\sqrt{d_v}}\|\mathbf{X}\|_2^2\|\tilde{\mathbf{W}}\|_2 \tag{56}$$

$$\leq \frac{\alpha}{\sqrt{d_v}}\|\mathbf{X}\|_2^2. \tag{57}$$

From our initialisation, we have $\|\tilde{\mathbf{W}}\|_2 = 1$ and by assumption, the spectral norm of $\mathbf{X}$ is bounded which allows us to conclude the result. □

---

[6]To see this, we note that we can write the Newton-Schultz iterates as a polynomial applied to the singular values of $\mathbf{M}$ (use the SVD decomposition) where the repeated application of the polynomial has a fixed point at 1 for values in the range $(0, \sqrt{3})$.

**Lemma F.3.** *There exists some constant $C > 0$ such that*

$$\|\exp[\mathbf{S}] - \mathbf{I}_{2d_v}\|_2 \leq C\alpha. \tag{58}$$

*Proof.* To control $\|\exp[\mathbf{S}] - \mathbf{I}_{2d_v}\|_2$, we note that

$$\frac{\partial}{\partial t} \exp(t\mathbf{D}) = \mathbf{D} \exp(t\mathbf{D}). \tag{59}$$

Therefore,

$$\int_0^1 [\mathbf{S}] \exp(t[\mathbf{S}]) dt = \exp[\mathbf{S}] - \mathbf{I}_{2d_v} \tag{60}$$

$$\implies \|\exp[\mathbf{S}] - \mathbf{I}_{2d_v}\|_2 \leq \int_0^1 \|[\mathbf{S}]\|_2 \|\exp(t[\mathbf{S}])\|_2 \tag{61}$$

$$\leq \|[\mathbf{S}]\|_2 \tag{62}$$

$$\leq \|\mathbf{B}\|_2^2 \|\mathbf{S}\|_2, \tag{63}$$

since $\exp(t[\mathbf{S}]) \in \mathrm{SO}(2d_v)$ for all $t \in [0,1]$. We can apply Lemmas F.1 and F.2 to conclude. $\square$

**Lemma F.4.** *There exists some constant $C > 0$ such that*

$$\left\| \frac{\partial \operatorname{vec} \mathbf{B}}{\partial \operatorname{vec} \mathbf{M}_0} \right\|_2 \leq C. \tag{64}$$

*Proof.* We first note that $\mathbf{M}_0$ has the form $\mathbf{M}/\beta$ where $\mathbf{M} = [\mathbf{X}\mathbf{W}^Q, \mathbf{X}\mathbf{W}^K]$ and $\beta = \|\mathbf{M}\|_F + \epsilon$ where $\epsilon > 0$. We have

$$\|\mathbf{M}_0\|_2 \leq \frac{1}{\epsilon} \|\mathbf{M}\|_2 \tag{65}$$

$$\leq \frac{1}{\epsilon} \sqrt{\|\mathbf{X}\mathbf{W}^Q\|_2^2 + \|\mathbf{X}\mathbf{W}^K\|_2^2} \tag{66}$$

$$\leq \frac{\sqrt{2}}{\epsilon} \|\mathbf{X}\|_2 \tag{67}$$

$$\leq C_1, \tag{68}$$

where $C_1 > 0$ is some constant and we use that our initialisation ensures $\|\mathbf{W}^Q\|_2 = \|\mathbf{W}^K\|_2 = 1$ and $\mathbf{X}$ has bounded spectral norm. Next, as the output $\mathbf{B} = \mathbf{M}_K$ of Newton-Schultz can be expressed as a polynomial, then the matrix differential of $\mathbf{B}$ can be expressed in terms of some polynomial of $\mathbf{M}_0$ which implies the Jacobian has bounded spectral norm. $\square$

**Lemma F.5.** *There exists some constant $C > 0$ such that*

$$\left\| \frac{\partial \operatorname{vec} \mathbf{M}_0}{\partial \operatorname{vec} \mathbf{X}} \right\|_2 \leq C. \tag{69}$$

*Proof.* We compute

$$\partial \mathbf{M}_0 = \frac{1}{\|\mathbf{M}\|_F + \epsilon} \partial \mathbf{M} + \mathbf{M} \partial \frac{1}{\|\mathbf{M}\|_F + \epsilon}, \tag{70}$$

and by using the fact that $\|\mathbf{M}\|_F = \operatorname{tr}(\mathbf{M}^\top \mathbf{M})$ and standard rules of matrix differentials, we have

$$\partial \frac{1}{\|\mathbf{M}\|_F + \epsilon} = -\frac{1}{(\|\mathbf{M}\|_F + \epsilon)^2} \frac{1}{2\|\mathbf{M}\|_F} \partial \operatorname{tr}(\mathbf{M}^\top \mathbf{M}) \tag{71}$$

$$= -\frac{1}{(\|\mathbf{M}\|_F + \epsilon)^2} \frac{1}{2\|\mathbf{M}\|_F} \operatorname{tr} \left( \partial \mathbf{M}^\top \mathbf{M} + \mathbf{M}^\top \partial \mathbf{M} \right) \tag{72}$$

$$= -\frac{1}{(\|\mathbf{M}\|_F + \epsilon)^2} \frac{1}{\|\mathbf{M}\|_F} \operatorname{tr} \mathbf{M}^\top \partial \mathbf{M}. \tag{73}$$

Therefore, we have

$$\partial \mathbf{M}_0 = \frac{1}{\|\mathbf{M}\|_F + \epsilon} \partial \mathbf{M} - \frac{1}{(\|\mathbf{M}\|_F + \epsilon)^2} \frac{1}{\|\mathbf{M}\|_F} \mathbf{M} \operatorname{tr} \mathbf{M}^\top \partial \mathbf{M}. \tag{74}$$

After vectorising this, we have

$$\partial \operatorname{vec} \mathbf{M}_0 = \frac{1}{\|\mathbf{M}\|_F + \epsilon} \partial \operatorname{vec} \mathbf{M} - \frac{1}{(\|\mathbf{M}\|_F + \epsilon)^2} \frac{1}{\|\mathbf{M}\|_F} \operatorname{vec} \mathbf{M} \operatorname{vec} \mathbf{M}^\top \partial \operatorname{vec} \mathbf{M} \tag{75}$$

$$\implies \frac{\partial \operatorname{vec} \mathbf{M}_0}{\partial \operatorname{vec} \mathbf{M}} = \frac{1}{\|\mathbf{M}\|_F + \epsilon} \mathbf{I} - \frac{1}{(\|\mathbf{M}\|_F + \epsilon)^2} \frac{1}{\|\mathbf{M}\|_F} \operatorname{vec} \mathbf{M} \operatorname{vec} \mathbf{M}^\top, \tag{76}$$

where we use the trivial fact that $\operatorname{tr} \mathbf{M}^\top \partial \mathbf{M} = \operatorname{vec} \mathbf{M}^\top \partial \operatorname{vec} \mathbf{M}$. It is easy to see that the Frobenius norm of the Jacobian is bounded as we assume $\|\mathbf{M}\|_2$ is bounded which implies $\|\mathbf{M}\|_F$ is also bounded (from the inequality $\|\mathbf{M}\|_2 \le \|\mathbf{M}\|_F \le \sqrt{\operatorname{rank}(\mathbf{M})}\|\mathbf{M}\|_2$). This allows us to conclude the result. $\qquad\square$

**Lemma F.6.** *There exists some constant $C > 0$ such that*

$$\left\| \frac{\partial \operatorname{vec} \mathbf{B}}{\partial \operatorname{vec} \mathbf{X}} \right\|_2 \le C. \tag{77}$$

*Proof.* This is a simple consequence from the chain rule and Lemmas F.4 and F.5. $\qquad\square$

**Lemma F.7.** *We have*

$$\left\| \frac{\partial \operatorname{vec} \exp[\mathbf{S}]}{\partial \operatorname{vec}[\mathbf{S}]} \right\|_2 \le 1. \tag{78}$$

*Proof.* From Higham (2008), we have that the vectorised Jacobian for the matrix exponential is given by

$$\frac{\partial \operatorname{vec} \exp[\mathbf{S}]}{\partial \operatorname{vec}[\mathbf{S}]} = \int_0^1 \exp(s[\mathbf{S}]^\top) \otimes \exp((1-s)[\mathbf{S}]) ds. \tag{79}$$

Therefore, we have

$$\left\| \frac{\partial \operatorname{vec} \exp[\mathbf{S}]}{\partial \operatorname{vec}[\mathbf{S}]} \right\|_2 \le \int_0^1 \|\exp(s[\mathbf{S}]^\top)\|_2 \|\exp((1-s)[\mathbf{S}])\|_2 ds = \int_0^1 1 ds = 1, \tag{80}$$

as the matrices $s[\mathbf{S}]^\top, (1-s)[\mathbf{S}]$ are skew-symmetric for all $s \in [0,1]$ hence the matrix exponential terms are always orthogonal. $\qquad\square$

**Lemma F.8.** *There exists some constant $C > 0$ such that*

$$\left\| \frac{\partial \operatorname{vec} \mathbf{S}}{\partial \operatorname{vec} \mathbf{X}} \right\|_2 \le C\alpha. \tag{81}$$

*Proof.* Using the above representation of $\mathbf{S}$, the differential of $\mathbf{S}$ is given by

$$\partial \mathbf{S} = \frac{\alpha}{\sqrt{d_v}} \left( \partial \mathbf{X} \tilde{\mathbf{W}} \mathbf{X}^\top + \mathbf{X} \partial \tilde{\mathbf{W}} \mathbf{X}^\top + \mathbf{X} \tilde{\mathbf{W}} \partial \mathbf{X}^\top \right). \tag{82}$$

Hence, the Jacobian has the form

$$\frac{\partial \operatorname{vec} \mathbf{S}}{\partial \operatorname{vec} \mathbf{X}} = \frac{\alpha}{\sqrt{d_v}} \left( (\mathbf{X} \tilde{\mathbf{W}}^\top \otimes \mathbf{I}_N) + (\mathbf{I}_N \otimes \mathbf{X} \tilde{\mathbf{W}}) \mathbf{K} \right), \tag{83}$$

where $\mathbf{K}$ denotes the communication matrix which is used to convert the vectorisation of a transposed matrix to the vectorisation of the original matrix (i.e. $\operatorname{vec} \mathbf{B}^\top = \mathbf{K} \operatorname{vec} \mathbf{B}$). An important property of the communication matrix is orthogonality. By our assumptions that $\|\mathbf{X}\|_2$ is bounded and $\|\tilde{\mathbf{W}}\|_2 = 1$ we can conclude the result. $\qquad\square$

We now turn to prove the theorem.

*Proof.* We start off with using the fact that $\|\mathbf{D}_1 \otimes \mathbf{D}_2\|_2 \leq \|\mathbf{D}_1\|_2 \|\mathbf{D}_2\|_2$. Therefore,

$$\|\mathbf{J}_1\|_2 \leq \|\mathbf{X}\|_2 \|\mathbf{W}^V \mathbf{W}^O\|_2 \|\mathbf{I}_N\|_2 \left\| \frac{\partial \operatorname{vec} \mathbf{A}(\mathbf{X})}{\partial \operatorname{vec} \mathbf{X}} \right\|_2 \tag{84}$$

$$\leq C_0 \left\| \frac{\partial \operatorname{vec} \mathbf{A}(\mathbf{X})}{\partial \operatorname{vec} \mathbf{X}} \right\|_2 \tag{85}$$

since by assumption, the spectral norm of $\mathbf{X}$ is bounded, we have initialised $\mathbf{W}^V \mathbf{W}^O$ so that its spectral norm is 1 and $\|\mathbf{I}_n\|_2 = 1$ for all $n$. This implies that we just need to control the Jacobian of our attention matrix. We proceed by computing the form of the differential of $\mathbf{A}(\mathbf{X})$:

$$\partial \mathbf{A}(\mathbf{X}) = \partial \mathbf{B}(\exp[\mathbf{S}] - \mathbf{I}_{2d_v})\mathbf{B}^\top + \mathbf{B}\partial(\exp[\mathbf{S}] - \mathbf{I}_{2d_v})\mathbf{B}^\top + \mathbf{B}(\exp[\mathbf{S}] - \mathbf{I}_{2d_v})\partial \mathbf{B}^\top. \tag{86}$$

We then have

$$\frac{\partial \operatorname{vec} \mathbf{A}(\mathbf{X})}{\partial \operatorname{vec} \mathbf{X}} = \mathbf{E}_1 + \mathbf{E}_2 + \mathbf{E}_3 \tag{87}$$

where

$$\mathbf{E}_1 = (\mathbf{B}(\exp[\mathbf{S}] - \mathbf{I}_{2d_v})^\top \otimes \mathbf{I}_N) \frac{\partial \operatorname{vec} \mathbf{B}}{\partial \operatorname{vec} \mathbf{X}} \tag{88}$$

$$\mathbf{E}_2 = (\mathbf{B} \otimes \mathbf{B}^\top) \frac{\partial \operatorname{vec}(\exp[\mathbf{S}] - \mathbf{I}_{2d_v})}{\partial \operatorname{vec} \mathbf{X}} \tag{89}$$

$$\mathbf{E}_3 = (\mathbf{I}_N \otimes \mathbf{B}(\exp[\mathbf{S}] - \mathbf{I}_{2d_v})\mathbf{K} \frac{\partial \operatorname{vec} \mathbf{B}}{\partial \operatorname{vec} \mathbf{X}}. \tag{90}$$

where $\mathbf{K}$ denotes the communication matrix.

1. For the first term, we can apply Lemmas F.1, F.3 and F.6 with submultiplicity to show that $\|\mathbf{E}_1\|_2 \leq O(\alpha)$.

2. For the second term, we have

$$\|\mathbf{E}_2\|_2 \leq \|\mathbf{B}\|_2^2 \left\| \frac{\partial \operatorname{vec}(\exp[\mathbf{S}] - \mathbf{I}_{2d_v})}{\partial \operatorname{vec} \mathbf{X}} \right\|_2. \tag{91}$$

By Lemma F.1, the spectral norm of the first term on the right-hand side is bounded by a constant. To control the second term, we note that

$$\frac{\partial \operatorname{vec}(\exp[\mathbf{S}] - \mathbf{I}_{2d_v})}{\partial \operatorname{vec} \mathbf{X}} = \frac{\partial \operatorname{vec} \exp[\mathbf{S}]}{\partial \operatorname{vec} \mathbf{X}} = \frac{\partial \operatorname{vec} \exp[\mathbf{S}]}{\partial \operatorname{vec}[\mathbf{S}]} \frac{\partial \operatorname{vec}[\mathbf{S}]}{\partial \operatorname{vec} \mathbf{X}}. \tag{92}$$

From Lemma F.7, the spectral norm of the first term on the right-hand side is bounded by 1. For the Jacobian of $[\mathbf{S}] = \mathbf{B}^\top \mathbf{S}\mathbf{B}$, we compute the differential as

$$\partial[\mathbf{S}] = \partial \mathbf{B}^\top \mathbf{S}\mathbf{B} + \mathbf{B}^\top \partial \mathbf{S}\mathbf{B} + \mathbf{B}^\top \mathbf{S}\partial \mathbf{B}. \tag{93}$$

Therefore, the Jacobian has the form

$$\frac{\partial \operatorname{vec}[\mathbf{S}]}{\partial \operatorname{vec} \mathbf{X}} = \hat{\mathbf{E}}_1 + \hat{\mathbf{E}}_2 + \hat{\mathbf{E}}_3, \tag{94}$$

where

$$\hat{\mathbf{E}}_1 = ((\mathbf{S}\mathbf{B})^\top \otimes \mathbf{I}_{2d_v})\mathbf{K} \frac{\partial \operatorname{vec} \mathbf{B}}{\partial \operatorname{vec} \mathbf{X}} \tag{95}$$

$$\hat{\mathbf{E}}_2 = (\mathbf{B}^\top \otimes \mathbf{B}^\top) \frac{\partial \operatorname{vec} \mathbf{S}}{\partial \operatorname{vec} \mathbf{X}} \tag{96}$$

$$\hat{\mathbf{E}}_3 = (\mathbf{I}_{2d_v} \otimes \mathbf{B}^\top \mathbf{S}) \frac{\partial \operatorname{vec} \mathbf{B}}{\partial \operatorname{vec} \mathbf{X}} \tag{97}$$

(a) For the first term, we can apply Lemmas F.2, F.1 and F.6 to conclude that $\|\hat{\mathbf{E}}_2\|_2 \leq O(\alpha)$.

(b) For the second term, we can apply Lemmas F.1 and F.8 to conclude that $\|\hat{\mathbf{E}}_2\|_2 \leq O(\alpha)$.

(c) For the third term, we can apply Lemmas F.1, F.2 and F.6 to conclude that $\|\hat{\mathbf{E}}_3\|_2 \leq O(\alpha)$.

Putting all of these results together with the triangle inequality, we can conclude that $\|\mathbf{E}_2\|_2 \leq O(\alpha)$.

3. For the third term, we can apply Lemmas F.1, F.3 and F.6 and the fact that $\mathbf{K}$ is orthogonal to conclude that $\|\mathbf{E}_3\|_2 \leq O(\alpha)$.

Putting all of these results together with the triangle inequality, we can conclude that $\|\mathbf{J}_1\|_2 \leq C\alpha$ for some $C > 0$. □

### F.7 PROOF OF THEOREM C.4

*Proof.* By our assumptions the non-zero singular values of $\mathbf{W}^V \mathbf{W}^O$ are 1. Further, it is a standard fact that the singular values of $\mathbf{J}_2 = (\mathbf{W}^V \mathbf{W}^O) \otimes \mathbf{A}(\mathbf{X})$ are the product of singular values of $\mathbf{W}^V \mathbf{W}^O$ and $\mathbf{A}$. Therefore, from the bound $|\sigma_i(\mathbf{A}(\mathbf{X})) - 1| \leq \delta$, the non-zero singular values of $\mathbf{J}_2$ satisfy

$$\sigma_i(\mathbf{J}_2) \in [1 - \delta, 1 + \delta]. \tag{98}$$

Then by Weyl's inequality and Theorem C.3, we have

$$|\sigma_i(\mathbf{J}) - \sigma_i(\mathbf{J}_2)| \leq \|\mathbf{J}_1\|_2 \leq C\alpha. \tag{99}$$

This implies that $\sigma_i(\mathbf{J}) \in [1 - \delta - C\alpha, 1 + \delta + C\alpha]$. Therefore, $\sigma_{\max}(\mathbf{J}) \leq 1 + \delta + C\alpha$ and $\sigma_{\min}(\mathbf{J}) \geq 1 - \delta - C\alpha$. If we take $\alpha$ such that $1 - \delta - C\alpha > 0$, we can conclude the result. □

