# OpenReview forum: "Orthogonal Self-Attention"
_ICLR.cc/2026/Workshop/GRaM — ICLR 2026 Workshop GRaM Poster_

### Official Review · Reviewer_Et36 · 2026-02-17
**Review of Orthogonal Self-Attention**

**Rating:** 8
**Confidence:** 3

**Review:**

The paper proposes orthogonal self-attention (OSA) as a replacement for softmax self-attention in transformers. Prior work argues that transformers without skip connections are worse from representation learning perspective as depth is not used, while removing them leads to training instability and rank collapse.

The approach uses the fact that the exponential of a skew-symmetric matrix is orthogonal. In the paper, the attention matrix is defined as the exponential of a matrix S, which is constructed from key and query matrices in a way that guarantees skew-symmetry. To overcome the complexity of computing the exponential, they take advantage of the low rank of S, reducing the computation to the exponential of a smaller matrix in another basis.

They perform a simple experiment on MNIST, showing that the performance of OSA matches the performance of ViTs, whereas performance of ViT without skip connections and layer norm is worse.

**strengths**

- Clear motivation: directly targets the rank collapse of skipless transformers.
- The solution feels more fundamental than ad-hoc stabilizers
- The computational aspects are taken care of by reduction to an $r \times r$ exponential.
- The presentation is very good

**weaknesses**

- The experiments are preliminary (understandable for a tiny paper)
- The experiments focus on loss/accuracy. It would be nice to see some plots of the effective rank of token representations across layers for the different models.

**Pmlr Suitability:**

NA

---

### Official Review · Reviewer_zpp4 · 2026-02-25

**Rating:** 6
**Confidence:** 4

**Review:**

In this paper author make an observation that softmax-attention (without no skip connections)  can induce rank collapse and poor Jacobian conditioning. To address this authors propose to impose orthogonality constraint on self-attention matrix by parameterizing it as an exp map of a skew-symmetric matrix S, where S is computed using Key and Query transforms. This approach preserves norms and Eigen-structure. It is an interesting idea and would benefit from more technical rigor. Some comments:

i) Paper use low-rank structure of S to reduce matrix exponential to an $r\times r$ computation using a learned basis which results in a linear scaling in sequence length. Basis construction is

ii) Jacobian conditioning relies on assumptions that are difficult to maintain at scale. Initialization on Stiefel, small \alpha so J_1 is a small perturbation, A is near-orthogonal. None of these are forced explicitly and training on deeper network can drift away.
the theoretical

iii) Training relies on a differentiable basis in every forward pass. As noted by authors QR can have exploding gradients, Newton-Schulz require normalization and enough iterations K to be close to orthonormal.  This is done per head per layer. There is no analysis on its cost with increasing depth.

iv) In Lemma F.8 differential includes a term that should be zero for a Jacobian wrt X. The term $X \partial \tilde{W} X^T$ looks like a typo?

v) Rank collapse theorem only holds true for OSA but a typical architecture includes MLP and other transformations. So rank preservation for full architecture is not established.

vi) In practice skip-connections are used together with self-attention. It is unclear what are immediate and practical benefit for developing scalable models. Besides, applicability is to non-causal transformers so it is unclear if proposed approach is of any benefit in training LLMs.

**Pmlr Suitability:**

NA

---

### Meta-Review · Area_Chair_Qc3e · 2026-02-27

**Decision:**

Accept

**Metareview:**

Reviewers were positive on this paper proposing an interesting idea to combat rank collapse by imposing orthogonality on the self-attention matrix.  Reviewers suggest increasing rigor and experimental scale for a full paper.

**Relevance To Proceedings:**

Tiny paper — does not apply

**Relevance To Workshop:**

Yes — suitable for GRaM

---

### Decision · Program_Chairs · 2026-03-02

Accept (Poster)